# Improving Enzymatic Saccharification of Peach Palm (*Bactris gasipaes*) Wastes via Biological Pretreatment with *Pleurotus ostreatus*

**DOI:** 10.3390/plants12152824

**Published:** 2023-07-31

**Authors:** Kamila de Cássia Spacki, Danielly Maria Paixão Novi, Verci Alves de Oliveira-Junior, Daniele Cocco Durigon, Fernanda Cristina Fraga, Luís Felipe Oliva dos Santos, Cristiane Vieira Helm, Edson Alves de Lima, Rosely Aparecida Peralta, Regina de Fátima Peralta Muniz Moreira, Rúbia Carvalho Gomes Corrêa, Adelar Bracht, Rosane Marina Peralta

**Affiliations:** 1Departamento de Bioquímica, Universidade Estadual de Maringá, Maringá 87020-900, Brazil; pg54413@uem.br (K.d.C.S.); ra122672@uem.br (D.M.P.N.); vercijuniorbio@gmail.com (V.A.d.O.-J.); luisfelipe.oliva11@gmail.com (L.F.O.d.S.); abracht@uem.br (A.B.); 2Departamento de Química, Universidade Federal de Santa Catarina, Florianópolis 88040-900, Brazil; daniele.cocco.durigon@gmail.com (D.C.D.); rosely.peralta@ufsc.br (R.A.P.); 3Departamento de Engenharia Química, Universidade Federal de Santa Catarina, Florianópolis 88040-900, Brazil; fe_fraga@live.com (F.C.F.); regina.moreira@ufsc.br (R.d.F.P.M.M.); 4Embrapa Florestas, Colombo 83411-000, Brazil; cristiane.helm@embrapa.br (C.V.H.); edson.lima@embrapa.br (E.A.d.L.); 5Programa de Pós-Graduação em Tecnologias Limpas, Instituto Cesumar de Ciência, Tecnologia e Inovação—ICETI, Universidade Cesumar—UNICESUMAR, Maringá 87050-900, Brazil; rubia.correa@unicesumar.edu.br

**Keywords:** circular economy, enzymes, renewable energy, upcycling

## Abstract

The white-rot fungus *Pleurotus ostreatus* was used for biological pretreatment of peach palm (*Bactris gasipaes*) lignocellulosic wastes. Non-treated and treated *B. gasipaes* inner sheaths and peel were submitted to hydrolysis using a commercial cellulase preparation from *T. reesei*. The amounts of total reducing sugars and glucose obtained from the 30 d-pretreated inner sheaths were seven and five times higher, respectively, than those obtained from the inner sheaths without pretreatment. No such improvement was found, however, in the pretreated *B. gasipaes* peels. Scanning electronic microscopy of the lignocellulosic fibers was performed to verify the structural changes caused by the biological pretreatments. Upon the biological pretreatment, the lignocellulosic structures of the inner sheaths were substantially modified, making them less ordered. The main features of the modifications were the detachment of the fibers, cell wall collapse and, in several cases, the formation of pores in the cell wall surfaces. The peel lignocellulosic fibers showed more ordered fibrils and no modification was observed after pre-treatment. In conclusion, a seven-fold increase in the enzymatic saccharification of the *Bactris gasipaes* inner sheath was observed after pre-treatment, while no improvement in enzymatic saccharification was observed in the *B. gasipaes* peel.

## 1. Introduction

The sustainable use of lignocellulosic biomass has been gaining ground and arousing the interest of researchers around the world, mainly because of its low cost and wide availability in nature [1]. These resources are excellent renewable raw materials as they are sources of organic compounds for growing microorganisms, favoring their use as substrates in the fermentative processes that can generate high value-added products, such as enzymes and biofuels [2]. Because they are obtained mainly from agricultural, agro-industrial, forestry and industrial residues, the correct use of these residues is desirable, since the accumulation and improper disposal of these materials can cause serious environmental problems [3].

*Bactris gasipaes* Kunth is a palm tree native to the Amazon region. It is popularly known as pupunha and its planting has recently expanded into the southeastern and southern Brazilian states (Figure 1). Its cultivation has two main purposes, namely palm heart and fruit (palm peach) production. In the Brazilian Amazon region, fruit production largely predominates, whereas in southern and southeastern Brazil, palm heart production is the almost exclusive reason for the cultivation of pupunha [4,5].

Brazil generates approximately USD 350 million annually from palm hearts, and it is estimated that the world market amounts to USD 500 million, still with considerable growth potential. In addition to being considered the largest producer and consumer of palm hearts, the country is the world’s largest exporter of the commodity, accounting for approximately 95% of all palm hearts consumed in the world. Of these, 90% are of extractive origin from açaí (*Euterpes oleracea* Mart.), native to the Amazon Forest, and from juçara (*Euterpes edulis* Mart.), native to the Atlantic Forest [4,5]. Although the economic index of the product is quite significant, the extraction of these species has contributed to endangering them, especially the juçara palm, which dies after the palm heart is harvested. Therefore, the rational use of other native palm trees to produce palm hearts, such as peach palms, has been promoted as a strategy to reduce the pressure of exploitation on *Euterpe edulis* Mart. (juçara) and *Euterpe oleracea* Mart. (açaí) [4,5].

A very great number of by-products are associated with the cultivation and processing of peach palm [6]. It has been estimated that nearly 84% of the total palm weight is accounted for by lignocellulosic wastes [7]. These wastes are environmentally and economically problematic and represent additional expenses for the producers, who must provide for their disposal. Ideally, proper disposal should fit into the upcycling concept, with a recycling and reuse approach, in order to close the product’s life cycle [8,9]. The transformation of this important biomass into marketable products represents an intelligent and promising way of dealing with otherwise useless waste. At the same time, it complies with the aims of sustainable development, which focuses on energy efficiency, food security and environment protection [10]. Feedstocks from lignocellulosic materials include cereal straw, bagasse, forest residues, and purpose-grown energy crops such as vegetative grasses and short rotation forests. Second-generation biofuels, such as bioethanol produced from lignocellulosic materials, have the potential to reduce greenhouse gas emissions and decrease dependence on fossil fuels. Furthermore, the production of these biofuels can promote economic and sustainable development in rural and urban areas. Research and development of technology for the production and use of these biofuels must be encouraged so that the full potential of these renewable resources can be taken advantage of and contribute to a more sustainable future.

To facilitate accessibility to the compounds of interest present in the resistant structure of lignocellulosic residues, it is necessary to apply a pre-treatment with the aim of increasing delignification and destroying the crystalline portions of cellulose for obtaining better yields of reducing sugars [11,12]. Several physical, chemical, physicochemical, and biological pretreatments have been used, including milling, microwaving, steam explosion, exposure to ionizing radiation, and treatment with acids, alkalis, oxidative agents, organic solvents and ionic liquids, or combinations thereof [12]. In general, these processes require high amounts of energy, excess water use and expensive chemicals. Considered a green treatment [13], biological pre-treatment is favored due to its mild conditions, non-polluting features and low cost and energy consumption [11,14]. Biological pre-treatment is an important method for processing lignocellulosic biomass and is carried out by the action of fungi, bacteria, or enzymes produced by these microorganisms [15,16,17]. The fungi identified as white rot are able to decompose various materials including corn cob, rice straw, wheat straw, sugarcane bagasse, and *Eucalyptus grandis* sawdust [18]. They are an excellent alternative for waste recovery, mainly because they can decompose carbohydrates and lignin present in most lignocellulosic materials [19]. One of them, *Pleurotus ostreatus*, is a cosmopolitan fungus, easy to cultivate in liquid and solid-state conditions, and able to produce many extracellular enzymes including ligninolytic ones, especially laccase (Lac, EC 1.10.3.2), and manganese peroxidase (EC 1.11.1.13). *P. ostreatus* and its ligninolytic enzymes have been used for the biotreatment of wastes and effluents [20], as well as for the bio-pretreatment of lignocellulose to produce second-generation biofuels [14]. Based on these considerations, in the present study, *P. ostreatus* was used in a biological pre-treatment aiming to establish the best conditions for the subsequent enzymatic hydrolysis of the cellulose from palm peach lignocellulosic wastes. Scanning electron microscopy, thermogravimetric analysis, and Fourier transform infrared spectroscopy were used to detect the main structural transformations caused by the *P*. *ostreatus* pre-treatment.

## 2. Results and Discussion

### 2.1. Growth of P. ostreatus and Enzyme Production

The biological pre-treatment consisted of growing *P. ostreatus* on *B. gasipaes* waste under stationary conditions (Figure 2). *P. ostreatus* can grow on different lignocellulosic residues thanks to its ability to produce different hydrolytic and oxidative enzymes capable of acting on lignocellulosic fibers [21]. This type of mycelial growth allowed the removal of fungal biomass and a direct obtainment of lignocellulosic waste. No significant differences in growth and mycelial biomass production were observed in cultures using the two lignocellulosic substrates (data not shown). Laccase was the major ligninolytic enzyme found in the culture filtrates over the whole incubation time (at least 20,000 U/L, Figure 2), while Mn peroxidase was detected at low levels, around 500–800 U/L. In general, solid-state fermentation is preferred for the cultivation of filamentous fungi because it provides a similar natural habitat for fungal growth and can be economically feasible to produce many biotechnological products [22]. SSF is also an interesting process to perform biological delignification. However, during the growth of fungi on SSF, the fungal hyphae penetrate and bind tightly to the substrate, which makes it difficult to separate the fungal biomass from the residual substrate. In this work, a high initial moisture content was used to reduce the porosity of the substrate and to limit oxygen transfer. Consequently, growth was restricted to the surface without reaching the underneath substrate mass (Figure 2). Biological pre-treatment of lignocellulosic wastes using white-rot fungi has been used to promote especially delignification. Some examples are the biological pre-treatments of *Eucalyptus grandis* sawdust with several white-rot fungi [14], fruit residues with *Pleurotus* sp [19], and tropical forage grass *(Panicum maximum)* with *Lentinus sajor-caju* [23]. Laccase appears to be the most important enzyme involved in the delignification process [23,24,25].

### 2.2. Chemical and Physical Characterization of Untreated and P. ostreatus-Pretreated Peach Pulp Waste

The cellulose, hemicellulose, and acid detergent lignin fractions of the cell wall constituents of the fibers were determined and are shown in Table 1. The compositions of both the inner sheaths and the peel were similar. The treatment tended to produce small diminutions in the cellulose, hemicellulose, and lignin contents. The lignin contents were the least affected by the treatment. The values in Table 1 do not differ from those found in other residues used in technological processes, such as sugarcane bagasse, corn straw and rice husks. Cellulose is almost always the most abundant component, with hemicellulose and lignin being the least abundant [26,27]. The extractives of peach palm inner sheath and peach palm peel were 35.12 ± 2.05 g/100 g of material and 30.20 ± 1.80 g/100 g of material, respectively. These extractive values are higher than those found in other lignocellulosic residues such as sugarcane bagasse (8.67–19.5%) [28], rice husk (1.82%) [29], and wheat straw (4.2–4.6%) [30].

The lignocellulosic fibers, with or without *P. ostreatus*-treatment, were further evaluated using three physical methods: Fourier transform infrared (FTIR) spectroscopy, scanning electron microscopy (SEM), and thermogravimetric analysis (TGA) The FTIR spectroscopy of untreated and pretreated samples is generally performed by focusing on the modifications in the bands corresponding to lignin, cellulose and hemicelluloses (Figure 3). These bands are 1515 cm^−1^ (aromatic skeletal vibrations in lignin), 1427 cm^−1^ (syringyl and guaiacyl condensed nuclei), 1098 cm^−1^ (crystalline cellulose), 1375 cm^−1^ (cellulose and hemicelluloses) and 898 cm^−1^ (amorphous cellulose). They are frequently used by authors to compare modifications in the lignocellulosic biomasses caused by different pre-treatments [14,31,32,33,34,35,36]. In the present study, however, only small and poorly defined decreases and increases were found in these bands upon biological pretreatment, although great modifications of different natures were found in the spectra. The main changes were observed at the wavenumbers of 1045, 1250, 1720, 2900 and 3335 cm^−1^, and they were different in the peel and inner sheaths. The peak at 1045 cm^−1^, for example, decreased in the peel but was substantially increased in the inner sheath samples. The same happened with the peaks at 1250, 1720, 2900 and 3335 cm^−1^. In the latter, especially, it was almost non-existent in the inner sheath samples before treatment and appeared very prominently after treatment. These spectral modifications are certainly difficult to interpret. They are not, in principle, due to modifications in the lignocellulosic fibers, but may be associated with modifications in other molecules that interact with the fibers such as soluble carbohydrates (glucose, fructose, and sucrose), and soluble polysaccharides (e.g., pectins), among several others. Specific analysis of these possible interactions is indispensable for a complete interpretation of these data.

Scanning electronic microscopy of the lignocellulosic fibers was performed to verify the structural changes caused by the biological pretreatments (Figure 4). The non-pretreated samples exhibited rigid and highly ordered fibrils (Figure 4A–C). The peel lignocellulosic fibers showed more ordered fibrils than the inner sheath fibers. Upon the biological pretreatment, the inner sheath structures were more extensively modified than the peel structures. The inner sheath fibers appear less ordered, with detachment of the fibers, cell wall collapse and with the formation of pores on the cell wall surfaces (Figure 4A,B). The formation of pores on the wall surfaces was more evident in the inner sheath material. Such microscopic alterations in the fibers have already been described for other kinds of treatments and have been generally considered to result from lignin removal [14,36,37,38]. The appearance of pores is usually considered as indicative of increases in the surface area of the cellulose available for enzyme attack [39]. On the other hand, comparison of the appearance of the peel fibers before and after pre-treatment reveals no significant modifications. (Figure 4C,D).

TGA/DTG analysis is an important tool that can be used to evaluate eventual changes in the composition of biomass submitted to pretreatment. It is also used to evaluate the content of hemicellulose, cellulose, and lignin because their decomposition temperature ranges are well known [40]. TGA/DTG analyses of untreated and pretreated samples are shown in Figure 5. As the pretreatment was not thermal, the TGA/DTG analyses were a complementary procedure for verifying whether there was mass loss and to certify that the cellulose content was not considerably affected in order to ascertain that the treatment was favorable for the intended purposes. There is a consensus that the thermogravimetric behavior of biomasses comprises three regions: the hemicellulose zone (245–290 °C), the cellulose zone (290–350 °C) and the lignin zone (350–500 °C) [41,42,43,44,45]. Figure 5 reveals that the 30-day treatment did not modify the temperatures at which the rates of weight loss for both cellulose and lignin were maximal. This is usually interpreted as an indication that no modifications in the molecular structure of these compounds took place.

### 2.3. Enzymatic Hydrolysis of B. gasipaes Inner Sheath and Peel with and without Biological Pretreatment

Non-treated and treated *B. gasipaes* inner sheath and peel were submitted to hydrolysis for 48 h using a commercial cellulase preparation from *T. reesei*. In all experiments, the same initial amounts of the lignocellulosic materials were incubated with the same amount of *T. reesei* cellulase under identical conditions. The results of the experiments in which the enzymatic production of free glucose and total reducing sugars was measured as a function of the treatment time are shown in Figure 6. At zero time (i.e., no treatment), the amount of total reducing sugars released by the cellulase from the peel clearly exceeded that released from the inner sheath by a factor of 2.1. However, no further increments in the hydrolysis products were found for the peel as a consequence of the treatment. For the inner sheath, by contrast, after 12 h of treatment, the release of total reducing sugars nearly doubled. In subsequent times, the release of total reducing sugars from the inner sheath increased exponentially with time. After 30 h of treatment, the total reducing sugars production increased by a factor of 7.2. In terms of free glucose, this represents an increment of 5.6-fold. The latter increment, if it results solely from cellulose hydrolysis, which is an unlikely hypothesis, represents 46.8% of the glucosyl units composing the polysaccharide backbone chains after the 30 days of *P. ostreatus* treatment. This value was calculated assuming that each µmol glucosyl moiety contributes 172 µg to the total weight of the polysaccharide. If a similar calculation is performed for the total polysaccharide content of the extract (cellulose plus hemicelluloses), the free glucose increment represents 31.8% of the whole original polysaccharide content. However, these figures correspond to the minimal yield after 48 h of incubation with the *T. reesei* cellulase preparation. The yield is probably higher as hydrolysis also produced reducing sugars with a higher molecular weight than glucose in addition to pentoses and reducing fragments derived from hemicelluloses.

The latter interpretation is corroborated by the thin-layer chromatographic separations shown in Figure 7. Lanes 5 and 7 are especially revealing: they show quite dense spots corresponding to free glucose (compared to glucose standard) and also to various di- and probably oligosaccharides, which in the assay react as reducing sugars. The density of the spots in lanes 2, 4 6 and 8, on the other hand, which correspond to the peel samples, is almost the same, an observation consistent with the absence of changes in free glucose and total reducing sugars shown in Figure 6.

### 2.4. Additional Relevant Aspects to Be Considered

As emphasized in the introduction, lignin poses a significant barrier to enzymatic hydrolysis. It can act as a physical barrier, preventing enzymes from efficiently accessing cellulose. Due to its rigid and hydrophobic nature, it can also hinder enzyme binding to cellulose, reducing the efficiency of the hydrolysis process. It can even bind non-specifically to enzymes, leading to their deactivation or denaturation [46]. All these phenomena decrease the abilities of the hydrolytic enzymes in breaking down the cellulose chains. Hemicellulose, on the other hand, affects cellulose hydrolysis in both positive and negative ways [47]. Its negative action occurs because it may reduce accessibility and act as an inhibitor by binding to the enzyme. Depending on their structure, however, hemicelluloses may also act as channels that facilitate enzymes accessing the cellulose chains within the lignocellulosic matrix. Furthermore, binding to the enzyme can also lead to partial hydrolysis of hemicellulose, releasing soluble and reducing sugars, including glucose, that can increase the saccharification yield. Optimization of the saccharification process thus requires a delicate balance between reducing the negative impacts of lignin and maximizing the beneficial effects of hemicellulose. In this respect, the treatment utilized in the present work seems to have been quite successful when applied to the *B. gasipaes* inner sheaths, but ineffective when applied to the peel.

The three methods used to evaluate changes in the lignocellulosic fibers showed that the pretreatment was very mild, resulting in small (inner sheath) or almost no changes (peel) in the fibers (Figure 3, Figure 4 and Figure 5). These results differ from those obtained previously when we used the same fungus, *P. ostreatus,* for the pretreatment of *Eucalyptus grandis* sawdust in solid-state cultivation [14]. Despite the use of two distinct lignocellulosic residues, in the pre-treatment carried out through solid-state cultivation, in addition to the action of oxidative enzymes that can act on the lignocellulosic fibers, there is a physical action produced by the hyphae in the substrate when the substrate is colonized by the fungus. Solid-state cultivation is largely used as a biological pretreatment of different lignocellulosic residues for improving the posterior enzymatic saccharification [48,49,50]. In the present work, the fungus grew solely on the surface and, therefore, this physical action on the substrate did not occur. The results obtained in the present work are similar to those obtained in a previous work by our group where the cryo-crushing technique was used. In this case, the alterations in the lignocellulosic fibers were very discreet, but sufficient to make them more accessible to attack by cellulolytic enzymes [36]. In fact, this is in agreement with a series of observations in which negative correlations between the lignin content and the degree of the enzymatic hydrolysis were observed. In these studies, pretreatments caused only a partial reduction in the lignin content, and the improvement of the saccharification could be more properly explained as a consequence of structural changes such as the formation of lignin droplets with increases in the pore volume [51] or a re-localization of lignin combined with a partial removal of hemicellulose [52,53]. Other examples are the observation that dilute acid and hydrothermal pretreatments caused some fragmentation of lignin with a slight delignification [54,55,56], but with a great improvement in saccharification. In the same way, saccharification of alamo switch grass (*Panicum virgatum*) was more improved by a pretreatment in which the accessibility to cellulose was facilitated rather than by an alkali pretreatment that resulted in the removal of a large amounts of lignin [51]. Still, after hydrothermal pretreatment of a *Populus* biomass, the glucose yield upon enzymatic hydrolysis improved in spite of a minimal removal of lignin [57]. Based on these and other similar observations [58,59], it thus seems reasonable to conclude that the lignin content per se does not affect recalcitrance significantly; rather, the integration of lignin and polysaccharides within the cell wall, and their associations with one another and with other wall components, are likely to play a more pronounced role in the biomass recalcitrance. All of these interpretations fit quite well into the results obtained in the present work: minimal or even the absence of lignin removal using the *P. ostreatus* treatment, but great increases in the access of the hydrolytic enzymes to cellulose and possibly other polysaccharides, as revealed by the saccharification yield. In this sense, the *P. ostreatus* treatment can be viewed as exerting a disrupting action on the integrated lignocellulosic structures, thus favoring the subsequent enzymatic attack. Evidence in favor of this disrupting action at a supramolecular level is evident in the images from electron microscopy.

It is important to emphasize at this point an additional advantage of stationary liquid cultures over solid-state cultures as a pretreatment of lignocellulosic biomass. This advantage derives from the fact that the complete removal of the mycelial biomass from liquid cultures is possible, whereas in solid-state cultures, this is practically impossible. As a consequence, it is also possible to obtain pretreated lignocellulose without contamination by the fungal biomass.

In the last few years, the use of peach palm lignocellulosic residues (inner sheaths and peel) within the context of the circular economy and upcycling has considerably increased. Some examples include the cultivation of the edible mushroom *Lentinula edodes* [60], the cultivation of *Trichoderma stromaticum* to produce important industrial enzymes such as xylanases [61] and cellulase [62], and the cultivation of *Ganoderma lucidum* for dye decolorization processes [63]. Peach palm wastes have also been used for obtaining important functional molecules such as xylo-oligosaccharide [6] and nanocellulose [7], antioxidant phenolics such as hydroxy benzoic acid, vanillic acid, gallic acid, and chlorogenic acid, and soluble sugars such as glucose, fructose, and sucrose [64].

## 3. Materials and Methods

### 3.1. Microorganism

*Pleurotus ostreatus* was obtained from the Basidiomycete Collection of the Laboratory of Biochemistry of Microorganisms and Food Science, Department of Biochemistry, State University of Maringá. The strain was maintained in the laboratory on potato dextrose agar (PDA) Petri dishes. The microorganism is deposited in the Basidiomycete Collection of the Laboratory of Biochemistry of Microorganisms and Food Science (Department of Biochemistry, State University of Maringá UEM) and is available to the scientific community.

### 3.2. Preparation and Characterization of Peach Palm Wastes (Inner Sheath and Peel)

The *B. gasipaes* wastes were obtained from Embrapa Florestas. The wastes were dried in the sunlight, milled to give a particle size of 2–3 mm thickness and used as raw materials in this study. Untreated and pretreated peach palm wastes (inner sheath and peel) were evaluated using the technique of acid detergent fiber (ADF) to obtain the percentages of cellulose, hemicellulose and lignin, and fiber neutral detergent (NDF) to obtain the percentage of lignocellulose [65]. For the determination of total soluble solids, 10 g of dry material plus 100 mL of distilled water were transferred into a 500 mL Erlenmeyer flask and maintained at 40 °C for 1 h under agitation of 100 rpm. The materials were filtered in a sintered glass crucible and dried in an oven (105 °C) until constant mass. The extractive contents were expressed on a dry basis.

### 3.3. Biological Pretreatment of Peach Palm Waste Using P. ostreatus

The biological pretreatment of peach palm waste with *P. ostreatus* was carried out in 250 mL Erlenmeyer flasks with 5 g of powder of peach palm waste (inner sheath and peel) plus 50 mL of mineral solution [66] supplemented with 0.5% glucose (*w/v*) and 0.1% (*w/v*) yeast extract. The materials were packed and autoclaved for 15 min at 121 °C. Four fully colonized PDA discs (10 mm diameter) were introduced into the culture media. The cultures were maintained for up to 30 days at 28 °C under static conditions in the dark. All of the cultures were grown in triplicate. The cultures were interrupted at different times. The mycelial biomass was removed with the aid of a spatula and the peach palm residue was filtered through gauze. The soluble material was centrifuged at 2000× *g* for 10 min and stored at −20 °C until use. This material was considered as the crude enzyme extract and was used for the determination of laccase and Mn peroxidase activities. The insoluble materials were washed three times with distilled water, and dried to a constant weight in a forced circulation oven at 40 °C. Control cultures (without inoculation of any fungus) were submitted to all procedures, including autoclaving and drying at 40 °C.

### 3.4. Laccase and Mn Peroxidase Assays

Laccase and Mn peroxidase were assayed as described previously [67]. Briefly, the laccase activity (EC1.10.3.2) was quantified using 2,2′-azino-bis (3-ethylbenzothiazoline-6-sulfonic) (ABTS) in 50 mM sodium acetate buffer pH 5.0 as substrate. ABTS oxidation was determined as the increase in absorbance at 420 nm (ε = 36 mM^−1^ cm^−1^). The Mn peroxidase activity (MnP; EC 1.11.1.13) was assayed spectrophotometrically by following the oxidation of 1 mM MnSO_4_ in 50 mM sodium malonate, pH 4.5, in the presence of 0.1 mM H_2_O_2_. Manganic ions, Mn^3+^, form a complex with malonate, which absorbs at 270 nm (*ε* = 11,590 M^−1^ cm^−1^). The enzyme activities were determined at 40 °C and expressed in international enzyme units per liter.

### 3.5. Characterization of Untreated and Treated Lignocellulose

The treated and untreated peach palm wastes were characterized using Fourier transform infrared (FTIR) spectroscopy, scanning electron microscopy (SEM), and thermogravimetric analysis (TGA). For FTIR spectroscopy, 2 mg of each dried sample was mixed with 200 mg KBr of spectroscopic grade and compressed into pellets at a pressure of about 1 MPa. The spectra were repeated three times with an average of 128 scans in the range between 400 cm^−1^ and 2000 cm^−1^_._ The spectral resolution was 2 cm^−1^. A Shimadzu SS-550 Superscan was used for performing electron microscopy (SEM). For the imaging procedures, the samples were sputter coated with gold layers. The thermal behavior of the biomass samples was evaluated by thermogravimetric analysis (TGA) using a Shimadzu DTG60/60H analyzer, under inert (N_2_-99.995%) flow (50 mL min^−1^). The analysis conditions were a heating rate = 10 °C min^−1^ up to 35 °C, holding for 60 min, and then a heating rate = 5 °C min^−1^ up to 700 °C and holding for 30 min. 

### 3.6. Saccharification of Untreated and Pre-Treated Peach Palm Waste

The fibers were subjected to enzymatic hydrolysis using cellulase from *Trichoderma reesei* ATCC 26921 (Sigma-Aldrich C 8546). An amount of 0.5 g of peach palm waste with or without biological pretreatment was added to a 50 mL Erlenmeyer flask. A volume of 10 mL of 50 mmol/L citrate buffer, pH 5.0, was added to each flask. The enzyme was added to obtain a final activity of 10 U/mL. The mixtures were maintained on a rotary shaker at 130 rpm at 37 °C for 48 h. After this time, samples were withdrawn and filtered under vacuum. The released reducing sugars were quantified using the 3,5 dinitrosalicylic acid (DNS) method [68]. The aldehyde group of reducing sugars converts 3,5-dinitrosalicylic acid to 3-amino-5-nitrosalicylic acid, which is the reduced form of DNS. The formation of 3-amino-5-nitrosalicylic acid results in a change in the amount of light absorbed at a wavelength of 540 nm. The absorbance measured using a spectrophotometer is directly proportional to the amount of reducing sugar. The pH of the reaction medium was tested for all situations. No changes were detected during the incubation time.

The reducing sugars produced were also evaluated using thin-layer chromatography. A volume of 2 μL of each sample was applied to a Silica Gel G/UV plate (Macherey-Nagel, Germany). A mixture of chloroform:acetic acid:water 6:7:1 was used as mobile phase. The run was repeated twice, and development was performed using 0.2% orcinol (w:v) in sulfuric acid:ethanol (5:95, *v/v*). Subsequently, the plate was placed in an oven at 100 °C for 5 min.

The released glucose was quantified using a commercial glucose oxidase-peroxidase method (Analisa). Glucose oxidase catalyzes the oxidation of β-D-glucose into D-glucono-δ-lactone with the concurrent release of hydrogen peroxide [69]. Subsequently, in a reaction catalyzed by peroxidase, the latter enters into a second reaction involving *p*-hydroxybenzoic acid and 4-aminoantipyrine with the quantitative formation of a quinonimine dye complex which is measured at 510 nm. Absorbance at this wavelength is directly proportional to the free glucose concentration.

### 3.7. Statistical Analysis

The GraphPad Prism software (version 8.0) was used. The results were expressed as the mean ± standard errors and were submitted to one-way variance analysis, followed by post hoc Student–Newman–Keuls testing. Significance of the differences between two means were assessed by Student’s *t* test. The 5% level was adopted as a criterion of statistical significance.

## 4. Conclusions

The conclusion is that biological pretreatment of the *Bactris gasipaes* inner sheath by *Pleurotus ostreatus* under stationary conditions can increase the enzymatic saccharification by up to seven-fold. No such improvement in enzymatic saccharification was found when the *B. gasipaes* peel was submitted to identical treatment. Treatment of the inner sheath with *P. ostreatus* thus offers favorable perspectives as a procedure for reducing the recalcitrance to hydrolysis of the *Bactris gasipaes* inner sheath. The absence of a similar improvement of the peel saccharification is somewhat puzzling. The similarity in composition suggests that this phenomenon could be a consequence of structural differences not yet identified by the hitherto applied methods. Clearly, new investigations are needed for a better understanding of these divergent observations. However, it is worth emphasizing that this is most probably the first article to use a ligninolytic fungus, *P. ostreatus*, as a biological pre-treatment of peach palm lignocellulosic wastes to obtain fermentable sugars.

## Figures and Tables

**Figure 1 plants-12-02824-f001:**
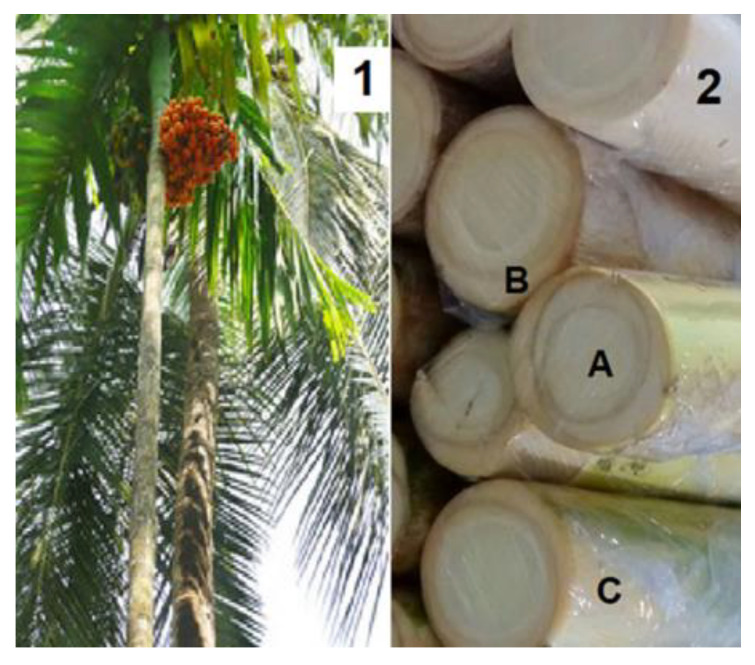
*Bactris gasipaes*. 1. Trees with fruits; 2. palm stalk. A: Peach palm heart (edible); B: inner sheath; C: peel.

**Figure 2 plants-12-02824-f002:**
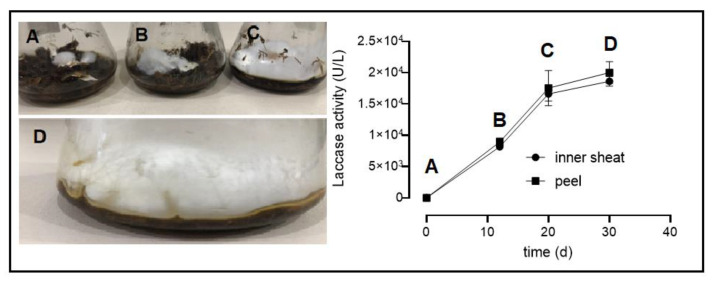
Pre-treatment of lignocellulosic residues of *Bactris gasipaes* with *P. ostreatus* under static conditions. The cultures developed using the *B. gasipaes* inner sheath and peel were maintained at 28 °C in the absence of light. Cultivation time: (**A**) 0 d; (**B**) 12 d; (**C**) 20 d; (**D**) 30 d. Laccase activities were evaluated in soluble materials.

**Figure 3 plants-12-02824-f003:**
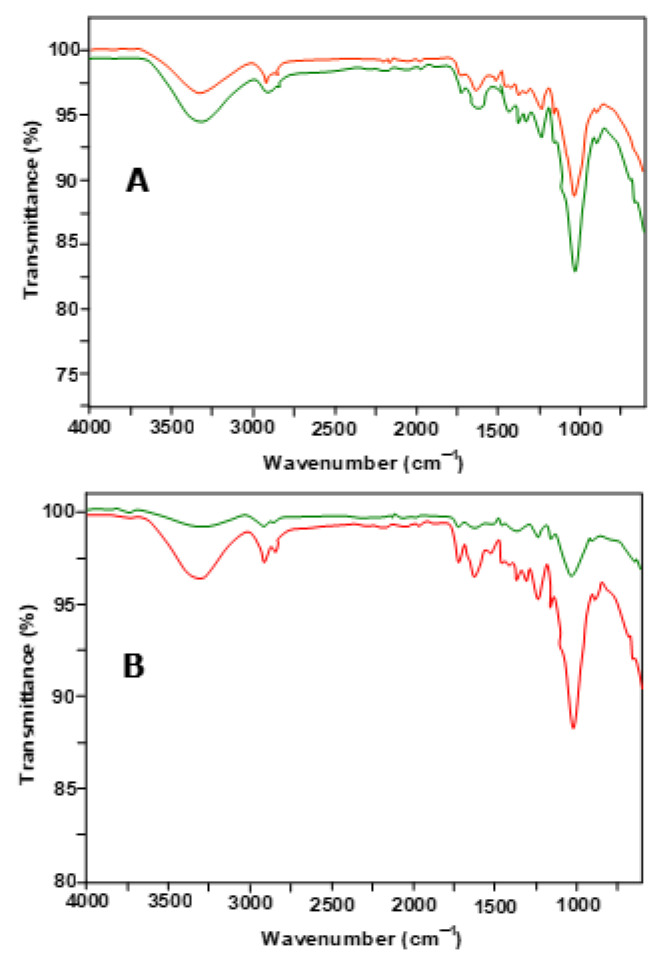
FTIR spectra of *B. gasipaes* peel (**A**) and inner sheaths (**B**) with (red line) and without (green line) biological pretreatment.

**Figure 4 plants-12-02824-f004:**
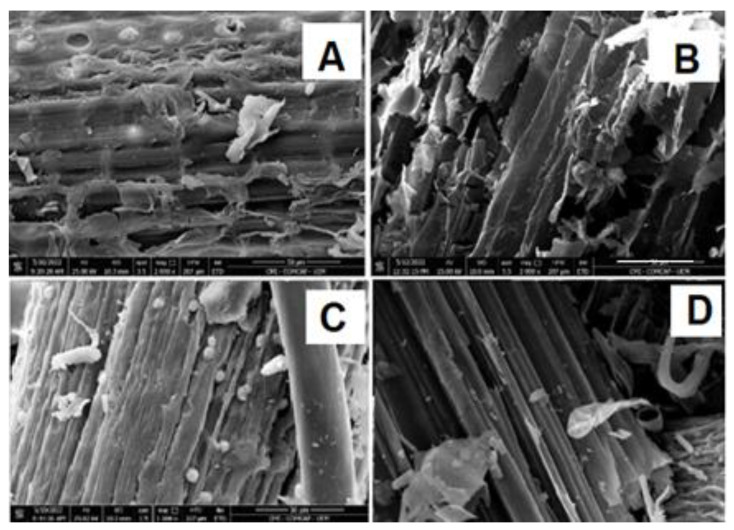
Scanning electron microscopy of the inner sheath control (**A**) and peel control (**C**), and the inner sheath (**B**) and peel (**D**) after 30 d of biological pretreatment with *P. ostreatus*. In the images, bars = 50 μm.

**Figure 5 plants-12-02824-f005:**
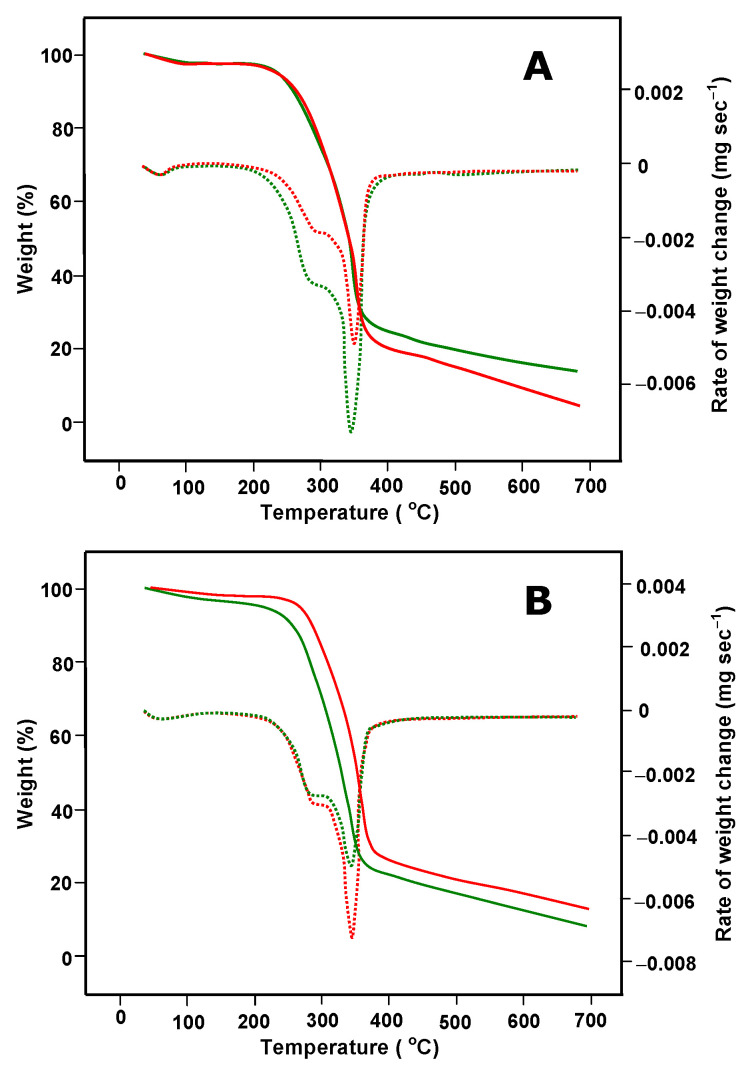
TGA analyses of untreated (green) and biologically pre-treated for 30 days (red) *B. gasipaes* peel (**A**) and inner sheath (**B**). The continuous lines represent the percent weight and the dotted lines indicate the corresponding rates of weight changes (derivative thermogravimetry, DTG).

**Figure 6 plants-12-02824-f006:**
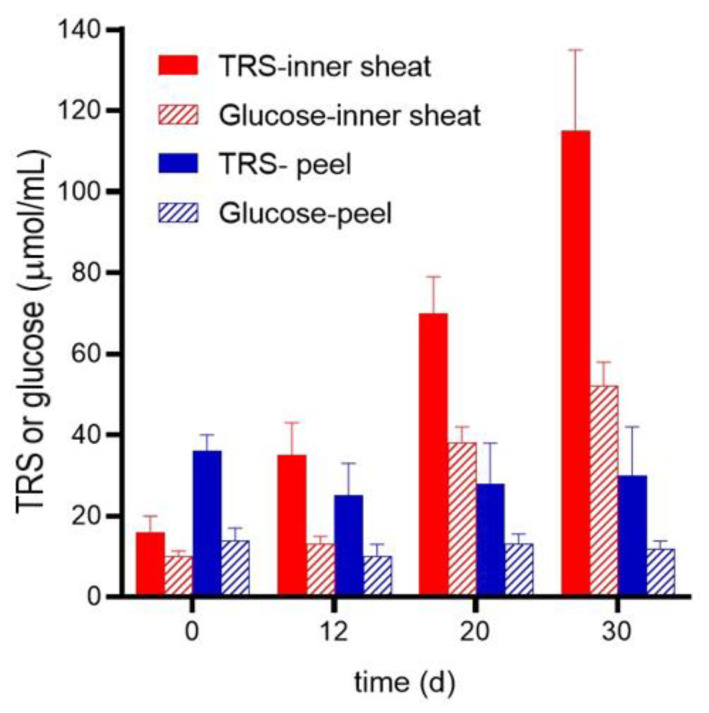
Enzymatic hydrolysis of *B. gasipaes* inner sheaths and peel with and without biological pretreatment. The incubation system (10 mL) contained 50 mmol/L citrate buffer (pH 5.0) and 0.5 g of lignocellulosic materials. Incubation was performed in Erlenmeyer flasks at 37 °C and under agitation (150 rpm). Aliquots of 0.3 mL were taken periodically for the determination of total reducing sugars (TRS) by means of the 3,5-dinitrosalicylic acid method and glucose by means of the glucose oxidase-peroxidase method. The experimental points are the means plus mean standard errors of 3 experiments.

**Figure 7 plants-12-02824-f007:**
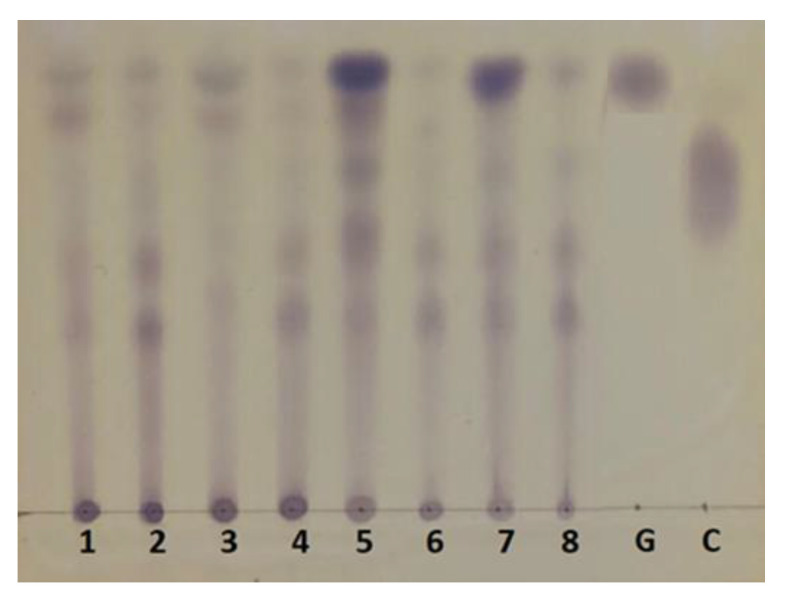
Thin-layer chromatography of the hydrolysis products of *B. gasipaes* inner sheath and peel using commercial cellulase. 1, 3, 5, and 7: Internal peach palm (inner sheath at zero, 12 d, 20 d and 30 d of biological pre-treatment); 2, 4, 6, 8: external peach palm (peel at zero, 12 d, 20 d and 30 d biological pre-treatment). Standards: G (glucose); C (cellobiose).

**Table 1 plants-12-02824-t001:** Contents of cellulose, hemicellulose, and lignin in *B. gasipaes* wastes (inner sheaths and peel) submitted or not to biological pretreatment with *Pleurotus ostreatus* during a period of 30 days.

Van Soest Fiber Analysis	Peach Palm Inner Sheaths (g/100 g Material)		Peach Palm Peel (g/100 g Material)	
	Untreated	Pretreated	Untreated	Pretreated
Cellulose	35.0 ± 5	30 ± 4	32.0 ± 4	29.0 ± 4
Hemicellulose	20.0 ± 3	14 ± 3	23.0 ± 5	18.3 ± 3
Acid detergent lignin	24.0 ± 4	21 ± 4	20.0 ± 5	19.2 ± 3

## Data Availability

The datasets supporting the conclusions of this article are included within the article.

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
