# Peer review of "Improving Enzymatic Saccharification of Peach Palm (Bactris gasipaes) Wastes via Biological Pretreatment with Pleurotus ostreatus"

_plants, 2023, doi:10.3390/plants12152824_

Round 1
Reviewer 1 Report
The article entitled “Improving enzymatic saccharification of peach palm (Bactris gasipaes) wastes by biological pretreatment with Pleurotus ostreatus” by Spacki et al. where the authors tried to show improved saccharification after biological pretreatment of peach palm inner sheath. The concept of the research is relevant, but more experimental studies are required to support and clarify the outcome. It is strange to see with a slight variation in the composition of peel and inner sheath of palm there is tremendous impact on the enzymatic saccharification.
1. Line 37-38: The authors mentioned enzymes as one of the by-products of lignocellulosic biomass which is not clear and the cited does not indicate the same. Change the sentence and make it clear as enzymes are not the byproducts of lignocelluloses.
2. ’Bactris gasipaes’ is sometimes in italics sometime not.
3. Compositional analysis of structural polysaccharides after biological pretreatment would be better to know the amount of lignocellulosic components left in the peel and inner sheath.
4. The Methods section is not clear and poorly explained.
5. Performing a chromatographic analysis to identify different monosaccharides after pretreatment and enzymatic hydrolysis is necessary to have the mass balance inaddition to reducing sugar and glucose assay.
6. XRD, NMR could be better to perform to understand why the peel fraction is highly resistant to enzymatic saccharification.
7. Language has to be thoroughly checked to make the concept clear.
Grammatical, spelling and extensive editing of language is required.
Author Response
Reviewer 1
The article entitled “Improving enzymatic saccharification of peach palm (Bactris gasipaes) wastes by biological pretreatment with Pleurotus ostreatus” by Spacki et al. where the authors tried to show improved saccharification after biological pretreatment of peach palm inner sheath. The concept of the research is relevant, but more experimental studies are required to support and clarify the outcome. It is strange to see with a slight variation in the composition of peel and inner sheath of palm there is tremendous impact on the enzymatic saccharification.
R.: Thank you for your analysis and suggestions. You will see that our contribution was substantially modified. A new table was introduced, more experimental details were given, and several aspects of the results were more extensively discussed. We are sure that our article was significantly improved by following your highly constructive suggestions.
- Line 37-38: The authors mentioned enzymes as one of the by-products of lignocellulosic biomass which is not clear and the cited does not indicate the same. Change the sentence and make it clear as enzymes are not the byproducts of lignocelluloses.
R: This is correct, the text was misleading and has been corrected.
2’Bactris gasipaes’ is sometimes in italics sometime not.
R: The whole text was screened for this and similar errors. Thank you.
3.Compositional analysis of structural polysaccharides after biological pretreatment would be better to know the amount of lignocellulosic components left in the peel and inner sheath.
R: This was done and the results are now presented in Table 1.
- The Methods section is not clear and poorly explained.
R: We have done our best in clarifying the methodology. Additions were made in several parts.
- Performing a chromatographic analysis to identify different monosaccharides after pretreatment and enzymatic hydrolysis is necessary to have the mass balance in addition to reducing sugar and glucose assay.
R: We believe that the dosage of total reducing sugars and glucose associated with TLC were sufficient to confirm the greater saccharification of the inner sheath material after pre-biological treatment
- XRD, NMR could be better to perform to understand why the peel fraction is highly resistant to enzymatic saccharification.
R: We agree with you. The two methods could help to highlight the differences between the two materials, but unfortunately, we did not have the equipment available to carry out the experiments at the time. We believe that the analyzes carried out by Fourier transform infrared (FTIR) spectroscopy, scanning electron microscopy (SEM), and thermogravimetric analysis (TGA), showed that although the two materials did not differ significantly in terms of lignin, cellulose and hemicellulose contents, a higher stiffness was observed in the peel (especially seen by microscopic analysis) and this may be the main reason why the biological method was not efficient to deconstruct the fiber and allow greater access of hydrolytic enzymes
- Language has to be thoroughly checked to make the concept clear.
R: As stated above we have scrutinized the whole text for errors and

Reviewer 2 Report
In this manuscript, the white-rot fungus Pleurotus ostreatus was used for biological pretreatment of peach palm (Bactris gasipaes) lignocellulosic wastes. A 7-fold increase in the enzymatic saccharification of the Bactris gasipaes inner sheath was observed after pre-treatment while no improvement of enzymatic saccharification was observed in the B. gasipaes peel. The manuscript conforms to the scope of the journal; the methods and results are relevant and accurate. Therefore, I think that it is worth publication.
In this manuscript, the white-rot fungus Pleurotus ostreatus was used for biological pretreatment of peach palm (Bactris gasipaes) lignocellulosic wastes. A 7-fold increase in the enzymatic saccharification of the Bactris gasipaes inner sheath was observed after pre-treatment while no improvement of enzymatic saccharification was observed in the B. gasipaes peel. The manuscript conforms to the scope of the journal; the methods and results are relevant and accurate. Therefore, I think that it is worth publication.
Author Response
Reviewer 2
In this manuscript, the white-rot fungus Pleurotus ostreatus was used for biological pretreatment of peach palm (Bactris gasipaes) lignocellulosic wastes. A 7-fold increase in the enzymatic saccharification of the Bactris gasipaes inner sheath was observed after pre-treatment while no improvement of enzymatic saccharification was observed in the B. gasipaes peel. The manuscript conforms to the scope of the journal; the methods and results are relevant and accurate. Therefore, I think that it is worth publication.
R: Thank you very much for encouraging observations.
Reviewer 3 Report
1. the other pretreatment methods for biorefinery should be referred in Introduction, the advantages of biological pretreatment should be emphasized.
2. The changes of chemical compositions after biological pretreatment should be added.
3. Plz add discussion about the effects of lignin and hemicellulose for enzymatic hydrolysis
4. How about the digestibility yields of cellulose of enzymatic hydrolysis yield?
none
Author Response
Reviewer 3
Thank you for your analysis and suggestions. You will see that our contribution was substantially modified. A new table was introduced, more experimental details were given and several aspects of the results were more extensively discussed. We are sure that our article was significantly improved by following your highly constructive suggestions.
- the other pretreatment methods for biorefinery should be referred in Introduction, the advantages of biological pretreatment should be emphasized.
R: Other pretreatments were added to the text
- The changes of chemical compositions after biological pretreatment should be added.
R: FTIR and thermogravimetric analyzes suggest that there were few changes in cellulose, hemicellulose, and lignin contents during the biological pretreatment. Even so, we analyzed the contents after 30 days of treatment. The data which are shown in newly introduced Table 1 confirm these assumptions.
- Plz add discussion about the effects of lignin and hemicellulose for enzymatic hydrolysis
R: A relatively long paragraph was introduced in subsection 2.4 for discussing this issue. Thank you for your suggestion.
- How about the digestibility yields of cellulose of enzymatic hydrolysis yield?
R: It is quite clear from figures 6 and 7 that the biological pre-treatment was efficient and increased saccharification by 7 times. It is also evident from figure 7 that glucose was the most produced sugar by enzymatic saccharification. These figures are in our opinion more important. Calculation of the cellulose yield in terms of the enzymatic hydrolysis is disturbed by two factors: (a) unknown amounts of di- and oligosaccharides are formed; (b) hemicelluloses may also have been partly hydrolyzed during the long time of incubation (48 hours). Even so we have made an attempt at estimating the yield in terms of the amounts of glucose that were effectively released from cellulose or from the total polysaccharides present in the incubation medium.

Round 2
Reviewer 1 Report
The authors tried to include more discussion points toexplain the differences in the enzymatic hydrolysis yields of inner sheath and peel of peach palm (Bactris gasipaes) wastes after biological pretreatment with Pleurotus ostreatus”. Compositional analysis of peach palm wastes were additionally added by incorporating the data in table 1. It would be better to mention the unit for % values mentioned in the table and in the lines 179-181. Minor language and format editing is required. Otherwise the article is in acceptable form and relevant to publish in the journal.
Minor language and format editing is required in introduction and results-discussion sections. Figures can be improved if possible.
Author Response
Reviewer 1
The authors tried to include more discussion points toexplain the differences in the enzymatic hydrolysis yields of inner sheath and peel of peach palm (Bactris gasipaes) wastes after biological pretreatment with Pleurotus ostreatus”. Compositional analysis of peach palm wastes were additionally added by incorporating the data in table 1. It would be better to mention the unit for % values mentioned in the table and in the lines 179-181. Minor language and format editing is required. Otherwise the article is in acceptable form and relevant to publish in the journal.
R: Thank you very much for the opportunity to improve the article. Your suggestion was accepted.
Reviewer 3 Report
none
Minor editing of English language required
Author Response
Reviewer 3.
Minor editing of English language required
R: Done